# Effects of the Phosphodiesterase 10A Inhibitor MR1916 on Alcohol Self-Administration and Striatal Gene Expression in Post-Chronic Intermittent Ethanol-Exposed Rats

**DOI:** 10.3390/cells13040321

**Published:** 2024-02-09

**Authors:** Luísa B. Bertotto, Dolly Lampson-Stixrud, Anushka Sinha, Nicki K. Rohani, Isabella Myer, Eric P. Zorrilla

**Affiliations:** Department of Molecular Medicine, The Scripps Research Institute, La Jolla, CA 92037, USA; lbertotto@scripps.edu (L.B.B.); dollyl@princeton.edu (D.L.-S.); sinha.anushka12@gmail.com (A.S.); nickirohani@vt.edu (N.K.R.); bellam1221@outlook.com (I.M.)

**Keywords:** ethanol or alcohol intake, dorsal striatum or nucleus accumbens, alcohol use disorder, medium spiny neuron, immediate-early gene expression, PDE10A

## Abstract

Alcohol use disorder (AUD) requires new neurobiological targets. Problematic drinking involves underactive indirect pathway medium spiny neurons (iMSNs) that subserve adaptive behavioral selection vs. overactive direct pathway MSNs (dMSNs) that promote drinking, with a shift from ventromedial to dorsolateral striatal (VMS, DLS) control of EtOH-related behavior. We hypothesized that inhibiting phosphodiesterase 10A (PDE10A), enriched in striatal MSNs, would reduce EtOH self-administration in rats with a history of chronic intermittent ethanol exposure. To test this, Wistar rats (*n* = 10/sex) with a history of chronic intermittent EtOH (CIE) vapor exposure received MR1916 (i.p., 0, 0.05, 0.1, 0.2, and 0.4 µmol/kg), a PDE10A inhibitor, before operant EtOH self-administration sessions. We determined whether MR1916 altered the expression of MSN markers (*Pde10a*, *Drd1*, *Drd2*, *Penk*, and *Tac1*) and immediate-early genes (IEG) (*Fos*, *Fosb*, Δ*Fosb*, and *Egr1*) in EtOH-naïve (*n* = 5–6/grp) and post-CIE (*n* = 6–8/grp) rats. MR1916 reduced the EtOH self-administration of high-drinking, post-CIE males, but increased it at a low, but not higher, doses, in females and low-drinking males. MR1916 increased *Egr1*, *Fos*, and *FosB* in the DLS, modulated by sex and alcohol history. MR1916 elicited dMSN vs. iMSN markers differently in ethanol-naïve vs. post-CIE rats. High-drinking, post-CIE males showed higher DLS *Drd1* and VMS IEG expression. Our results implicate a role and potential striatal bases of PDE10A inhibitors to influence post-dependent drinking.

## 1. Introduction

Annually, approximately 13% of Americans suffer from alcohol use disorder (AUD) [1], a chronic disorder characterized by an inability to control ethanol (EtOH) use despite adverse social, occupational, and health outcomes [2]. AUD disables and increases mortality risk by nearly 6-fold [1,3]. Only about 1 in 5 individuals receive treatment [1]; one-quarter relapse within 3 years [4]. New therapeutic targets for AUD are thus of great societal interest.

Phosphodiesterases (PDEs) are a family of enzymes that regulate diverse cellular responses by hydrolyzing cyclic adenosine 3′, 5′—monophosphate (cAMP) and cyclic guanosine 3′, 5′—monophosphate (cGMP) [5,6] to inactive forms. Previous studies have linked cAMP and cGMP system molecules, including synthetic adenylyl (AC) and guanylyl (GC) cyclases and inactivating PDEs, to EtOH-related behaviors [7,8,9]. Relative to the other 20 PDE genes, the PDE10A isoform shows the highest expression in the caudate and putamen of the dorsal striatum and somewhat lower levels in the ventral striatum, including the nucleus accumbens (NAc) [10,11]. PDE10A is differentially expressed in medium spiny neurons (MSNs), which comprise ~95% of striatal neurons, and is putatively the main PDE isoform that physiologically controls MSN cAMP and cGMP levels in the dorsal striatum [12,13,14]. In rats, chronic EtOH access leads to regional changes in the striatal PDE10A binding potential in relation to EtOH intake and preference [15]. Interestingly, chronic EtOH use also leads to a shift in control over EtOH-seeking behaviors from the ventromedial (VMS) to dorsolateral striatum (DLS), where PDE10A is differentially expressed; this shift corresponds with a transition from goal-directed to compulsive- and habit-like drinking [16]. Given the differential localization of PDE10A in dorsal striatal MSNs that subserve compulsive EtOH-seeking and reinforcement and that undergo EtOH adaptation, we and others hypothesized a role for PDE10A in AUD neurobiology and treatment [7,8,9,10,11,12,13,14,15]. Accordingly, a tool-like selective PDE10A inhibitor, TP-10, reduced EtOH self-administration in male rats in acute EtOH withdrawal, genetic alcohol preference, and post-traumatic models; effective doses did not alter EtOH pharmacokinetics, motor function, or social interaction [8]. In the present study, we hypothesized that the systemic administration of MR1916, a drug-like PDE10A inhibitor that reached human clinical trials (*IC*_50_ = 0.02 nM, 1000-fold selectivity vs. other PDEs) for other indications [17], would decrease EtOH self-administration in post-dependent rats that still drink pharmacologically relevant EtOH levels weeks after the cessation of EtOH vapor exposure. To avoid nonspecific motor effects, a bar test [18] was performed to identify (and stay below) doses of MR1916 that elicit cataleptic-like behavior.

We also determined whether MR1916 induced immediate-early gene markers of striatal neuronal activation. Previously studied PDE10A inhibitors reportedly increase immediate-early gene expression in striatal MSNs [19,20,21]. Importantly, IEGs are heterogeneous in their inducer and functional roles [19,21,22], and the sensitivity of individual IEGs as markers of MR1916 action at doses that reduce EtOH self-administration is unknown. Thus, we studied MR1916 effects on the striatal expression of key IEGs that are putatively responsive to cAMP/cGMP cascades (*Egr1*, *Fos*, *Fosb*, Δ*Fosb*) [23,24,25,26,27,28,29,30,31,32,33] in EtOH-naïve and post-dependent rats. Given the distribution of PDE10A and previous work with other PDE10A inhibitors [19,20,21,22,34], we hypothesized that effective MR1916 doses would differentially increase IEG expression in the DLS (vs. NAc), with the greatest induction of *Egr1*.

Importantly, the activation of direct vs. indirect MSN pathways (dMSN, iMSN) has different effects on action selection and reward processing [35]. Heuristically, the activation of dMSNs promotes appetitive behavior, whereas iMSN activation stops behavior or promotes avoidance [36]. Relatively underactive iMSNs or overactive dMSNs putatively lead to increased and compulsive-like EtOH use [15,37,38,39,40]. The MSN pathways express different molecular markers; dMSNs, which extend to the basal ganglia output regions, differentially express dopamine D1 receptors (*Drd1*), and substance P (*Tac1*); iMSNs, which indirectly influence the basal ganglia output via the globus pallidus externa (GPe) and subthalamic nucleus, express dopamine D2 receptors (*Drd2*), and proenkephalin (*Penk*) [35]. In contrast, *Pde10a* is a pan-MSN marker [8]. Previously studied PDE10A inhibitors vary in their relative activation of dMSNs vs. iMSNs, as inferred from the expression of these markers [21,41]. This variation is hypothesized to reflect differences in the enzyme off-rate [41], whereby slow off-rate compounds putatively elicit a dMSN-biased activation and fast off-rate compounds elicit a balanced or greater iMSN activation [41]. Here, we tested the hypothesis that MR1916 differentially induced the expression of indirect (vs. direct) MSN markers at doses that reduced EtOH self-administration.

Finally, because prior studies have reported in vitro sex differences in the effects of PDE10A inhibitors on D1- and D2-responsive striatal neurons [42] and our previous study of TP-10 effects on EtOH self-administration only involved males [15], we tested the hypothesis that there would be greater behavioral and molecular effects of MR1916 in males than females.

## 2. Materials and Methods

### 2.1. Subjects

Adult (3 months old) male and female, group-housed (2–3/cage) Wistar rats (Charles River, Inc., Wilmington, MA, USA) were studied for catalepsy (*n* = 32/sex), post-CIE self-administration (*n* = 10/sex), and striatal gene expression (*n* = 21/sex). Rats were maintained in a temperature- (20–22 °C) and humidity-controlled (30–70%) vivarium under a reverse light cycle (lights off at 8 a.m./on at 8 p.m.). Food (Product #7012, Teklad Diets-Envigo, Madison, WI, USA) and tap water were available ad libitum. All procedures conformed to the National Institutes for Health Guide for the Care and Use of Animals and were approved by the Institutional Animal Care and Use Committee of Scripps Research Institute.

### 2.2. Bar Test

A bar test was performed. MR1916 has good bioavailability and brain penetration in rats [43]; non-cataleptic doses previously increased striatal cAMP and cGMP levels and improved cognitive performance and motor control in models of schizophrenia [44] and dyskinesias [43,45]. MR1916 (synthesized by Mochida Pharmaceuticals) (0, 0.1, 0.4, and 0.8 µmol/kg) free base, dissolved in 10% (*w*/*v*) 2 -hydroxypropyl-β-cyclodextrin (HPBC, Thermo Fisher Scientific, Waltham, MA) in Nanopure water was injected intraperitoneally (1 mg/mL). After 60 min, the rat’s forepaws were placed on a horizontal metal cylindrical bar rod ~10 cm high; hind paws were maintained on the metal floor. The latency it took for the rat to move both of its forepaws off the bar (in any manner) was recorded. Each subject received 3 trials with an intertrial interval of 10–20 min in a dark procedure room under red light.

### 2.3. Chronic Intermittent EtOH (CIE) Vapor Exposure

We used the standard alcohol inhalation method of The Scripps Research Institute Alcohol Research Center as a model of post-dependence [46]. Rats received chronic intermittent EtOH (CIE) vapor exposure via cage housing in specialized airtight chambers with regulated airflow and intermittent exposure to EtOH vapor (cycles of 14 h on and 10 h off), volatilized by heat. Post-session tail blood EtOH levels were measured weekly, and the flow rate was regulated to result in EtOH levels between 135–250 mg/dL, per gas chromatographic analysis (7820A, Agilent Technologies, Santa Clara, CA, USA) with a headspace sampler (7697A, Agilent Technologies). The median blood EtOH level maintained was 168.6 mg/dL (169.3 mg/dL males, 167.9 mg/dL females) with a semi-interquartile range of 33.5 mg/dL. The total duration of CIE exposure was 10–14 weeks.

### 2.4. EtOH vs. Water Operant Self-Administration

During at least the final two weeks of EtOH vapor exposure, rats received EtOH self-administration sessions during cessation of vapor exposure, which allowed them to learn negative reinforcing effects of EtOH during acute withdrawal. For daily self-administration acquisition, rats first received 23 h sessions (with chow available) in which responses at one lever delivered 0.1 mL water (fixed-ratio 1 (FR1)). Rats were first pair-trained and then individually trained to a criterion of 100 responses (1–2 days each). Individual rats then received 2 h sessions during which responses at the other lever delivered 0.1 mL 10% *w*/*v* EtOH (FR1) to a criterion of 15 responses, followed by 1 h FR1 sessions for EtOH to a criterion of 10 responses. Thereafter, rats received 2 lever choice sessions for EtOH (FR1) vs. water (FR3) during 2–3 weekly self-administration sessions until the start of testing 2–18 weeks after cessation of vapor exposure, by which time post-CIE EtOH responding had stabilized (daily intake and preference varied <20% from moving weekly average). Differences in effects of MR1916 on EtOH self-administration were not seen as a function of shorter (2 or 10 weeks) vs. longer (18 weeks) post-CIE intervals. The different ratio requirements were designed to yield similar rates of responding, with stable mean pre-testing preference (vs. water) for the EtOH reinforcer of 80.8% and 87.5% for females and males, respectively.

For testing, MR1916 treatment (0, 0.05, 0.1, 0.2, and 0.4 µmol/kg, dissolved in 10% (*w*/*v*) HPBC) was given in a within-subject, modified Latin square design with dose order counter-balanced. Sixty minutes after injections, subjects performed 1 h self-administration sessions of 10% (*w*/*v*) EtOH vs. water. Sessions were performed every 2–3 days, and test days were separated by at least one treatment-free session. An experimental timeline of the experiments described in Section 2.3 and Section 2.4 is presented in Figure 1.

### 2.5. Striatal Punch Collection in Naïve and Post-CIE Rats

Naïve rats received a non-cataleptic dose of MR1916 (0 or 0.1 µmol/kg, in 10% (*w*/*v*) HPBC). To study effects during post-dependence, separate rats received CIE and 2-lever EtOH self-administration sessions as described above until responding stabilized, 2–3 weeks after cessation of 10–12 weeks of CIE. Based on self-administration results, post-CIE rats received MR1916 (0, 0.05, or 0.4 µmol/kg, in 10% (*w*/*v*) HPBC). All rats were euthanized 90 min post-injection via rapid decapitation under isoflurane anesthesia (with no self-administration session). The lower dose was chosen because it did not reduce EtOH self-administration and even increased intake or preference, whereas the higher dose reduced EtOH self-administration in male rats. The dissected brain was sectioned using a 2 mm coronal slice wire-matrix, and punches of the DLS and NAc were taken using a 13-gauge blunt needle on an ice-cold stage. The tissue was transferred to a 1.5 mL centrifuge tube, flash-frozen in liquid nitrogen, maintained on dry ice post-dissection, and then placed in a −80 °C freezer until subsequent processing.

### 2.6. RNA Isolation, Reverse Transcription, and qPCR Analysis

RNA isolation was performed using QIAzol lysis reagent (Qiagen, Inc., Valencia, CA, USA) with shearing disruption using a discardable pestle and pestle motor, following the manufacturer’s instructions. RNA quantity and purity were determined via Nanodrop 2000c (Thermo Fisher Scientific, Waltham, MA, USA). Extracted total RNA (*M* ± SEM: 64.8 ng ± 3.3) was ezDNase-treated and reverse-transcribed using SuperScript IV (SSIV) reverse transcriptase (RT) VILO kit (Thermo Fisher Scientific, Waltham, MA, USA), following the manufacturer’s instructions. Gene expression levels were determined through a quantitative polymerase chain reaction (qPCR), performed using PowerTrack™ SYBR Green Master Mix (Thermo Fisher Scientific, Waltham, MA, USA), in a CFX 384 Real-Time System (Bio-Rad, Hercules, CA, USA) thermocycler. Cycling conditions were as follows: 95 °C denaturation temperature for 15 s; 60.3 °C annealing temperature for 15 s, and 72 °C extension temperature for 15 s. Primers were obtained from Integrated DNA Technologies (Coralville, IA, USA). Appendix A shows the primer sequences used in this study. The ΔΔCt method, utilizing the geometric averaged Ct of two housekeeping reference genes (*Ywhaz* and *Ppia*), was used to calculate the relative fold change of gene expression. References are included for primers retrieved from the literature. Other primers were designed in house. The expected single PCR products were confirmed by melt curve analysis.

### 2.7. Statistical Analysis

Catalepsy bar test latencies were log-transformed and analyzed using a two-way ANOVA with trial as a repeated measure and Dose as a between-subject factor.

To identify rats that still drank pharmacologically relevant (“high”) levels during post-dependence, a threshold of 0.6 g/kg was used to define “high-drinking” vs. “low-drinking” rats based on their untreated performance when stable responding was attained post cessation of CIE. This threshold of 0.6 g/kg to define high- vs. low-drinking rats also has been used by other groups [16,47] and was consistent with frequency histograms of male rat intake before MR1916 injections.

EtOH preference ratios for each subject were calculated as 100× the number of EtOH reinforcers earned divided by the total reinforcers earned (EtOH + water reinforcers). Data were analyzed using separate two-way analysis of variance for each sex with High- vs. Low-Drinking as a between-subject factor and Dose as a within-subject factor; linear and quadratic Dose contrasts were performed to identify dose-related effects. Following significant ANOVA effects, post-hoc LSD tests were performed to identify significant pairwise differences vs. vehicle control.

Gene expression analysis was performed using *z*-scores of the ΔΔCt’s (multiplied by −1 to represent directionality), standardized to values of female vehicle controls. Potential outliers were defined a) using the Dixon’s Q-test [48,49] (on raw Ct data) and b) as *z*-scored values with high studentized residual scores (≥|3.0|) that had undue leverage (>2-fold the jackknifed mean leverage of other samples) [50] or influence (Cook’s *D* > the 50th percentile of the *F*-distribution, *F*[Cook’s *D*](*k* + 1,*n* − *k* − 1), where *k* = number of predictor variables and *n* = total samples) [50]. Outliers and missing values (5 of 1092 scores in the post-CIE qPCR results, 0.46% of data) were replaced using multiple imputation [51] as the average of 10 independent estimates in SPSS. Both low-drinking males that received the higher dose of MR1916 were outliers by these criteria for *Drd2* and *Penk* expression in the DLS, making imputation and subgroup analysis for these markers impossible.

Gene expression data were analyzed by ANOVA with drug, sex, and high- vs. low-drinking as between-subject factors; subject cohort was a covariate. Following significant tests, post hoc pairwise comparisons used LSD tests for data with equal variance and Dunnett’s T3 for data with unequal variance. To help interpret magnitude of IEG induction within each sex, MR1916-induced increases in expression were also calculated as z-score differences. Baseline used in the Pearson correlation analysis was calculated as the average of the last 3 days of EtOH intake prior to MR1916 exposure, when post-CIE EtOH responding was stable. Effects and comparisons of *p* < 0.05 were defined as statistically significant. *F*-statistics and *dfs* for all significant omnibus ANOVA tests, as well as Dose contrast effects, are summarized in Appendix A.

## 3. Results

### 3.1. Catalepsy

Appendix A shows the effects of MR1916 treatment on the mean duration of catalepsy by female and male rats in the bar test. A Dose main effect (*p* = 0.032, Appendix A), followed by pairwise comparisons, showed that rats injected with 0.8 µmol/kg MR1916 had a significantly greater catalepsy time than vehicle-treated subjects (*p* = 0.029). The 0.4 µmol/kg and lower doses of MR1916 did not significantly increase the catalepsy duration over vehicle controls and were studied further in EtOH self-administration models.

### 3.2. Post-CIE Operant EtOH Self-Administration

#### 3.2.1. EtOH Intake

MR1916 pretreatment yielded a significant interaction of Dose X Drinking level (*p* = 0.036), in the context of a main effect of Drinking level (*p* = 0.02), as well as an overall quadratic Dose effect (*p* = 0.004) on the self-administered EtOH intake of post-CIE male and females analyzed together. Furthermore, MR1916 pretreatment led to a linear Dose effect (*p* = 0.03, Figure 2A), as well as a Dose X Drinking Level interaction (*p* = 0.012, Figure 2A) on the self-administered EtOH intake (g/kg) in post-CIE male rats analyzed separately. Follow-up pairwise comparisons to interpret the Dose X Drinking Level interaction showed that the highest dose of MR1916 significantly reduced the intake of high-drinking males vs. vehicle conditions (*p* = 0.011, Figure 2A, left panel). In contrast, lower doses of 0.05 µmol/kg and 0.2 µmol/kg significantly increased the intake of low-drinking males (*p* = 0.037 and *p* = 0.032, respectively, Figure 2A, right panel), resulting in a significant quadratic Dose contrast (*p* = 0.027).

In females, unlike in high-drinking males, an overall quadratic Dose contrast effect of MR1916 was observed (*p* = 0.023, Figure 2B). However, no pairwise comparisons versus vehicle were statistically significant, suggesting no clear dose-dependency to increase or decrease drinking in females.

#### 3.2.2. EtOH Preference

MR1916 pretreatment yielded a significant quadratic Dose contrast (*p* = 0.023) on the EtOH reinforcer preference of male and female post-CIE rats in the context of a main effect of Sex (*p* = 0.039). In contrast, MR1916 pretreatment did not significantly alter the EtOH reinforcer preference in male rats alone. However, MR1916 increased the EtOH reinforcer preference in post-CIE female rats alone per a quadratic Dose contrast (*p* = 0.001, Figure 3). Accordingly, a significant quadratic Dose contrast was seen in both high- (*p* = 0.034, Figure 3B, left panel) and low- (*p* = 0.031, Figure 3B, right panel) drinking females considered separately. Pairwise comparisons showed that the lowest dose (0.05 µmol/kg) of MR1916 significantly increased the EtOH reinforcer preference in low-drinking female rats, in comparison to vehicle conditions (*p* = 0.013, Figure 3B, right panel).

#### 3.2.3. Water Intake

However, MR1916 did not yield a significant ANOVA effect involving Dose on water intake in either male or female rats (Appendix AA,B); a significant main effect of Sex was seen (*p* = 0.014).

### 3.3. Gene Expression in Naïve Rats

#### 3.3.1. MSN Marker Expression

In the DLS, MR1916 exposure significantly increased the expression of *Drd1* (*p* = 0.038, Figure 4B), *Drd2* (*p* = 0.03, Figure 4C), and *Pde10a* (*p* = 0.03, Figure 4E) in EtOH-naïve female and male rats. MR1916 descriptively induced increases in *Drd1* (avg *z*-score increases = 1.11, and 0.56 for males and females) and *Drd2* (avg *z*-score increases = 0.67; and 1.05 for males and females) expression. Sex effects showed that the expression of *Pde10a* was significantly higher in naïve male vs. female rats irrespective of treatment (*p* = 0.005, Figure 4E).

In the VMS, MR1916 did not significantly alter the MSN marker expression. Sex effects showed that, irrespective of treatment, naïve male rats had a significantly higher expression of *Tac1* (*p* = 0.033), *Drd1* (*p* = 0.039), *Drd2* (*p* < 0.001), and *Pde10a* (*p* < 0.000) (Figure 5A–C,E) vs. female naïve rats.

#### 3.3.2. IEG Expression

As seen in Table 1 and Figure 6, the Dose main effects showed that MR1916 significantly increased the expression of *Egr1* (*p* < 0.001, Figure 6A), *FosB* (*p* = 0.002, Figure 6C), and Δ*FosB* (*p* = 0.013, Figure 6D) in naïve rats. MR1916 induced descriptively higher increases in male compared to female rats for *Egr1*, *Fos*, *FosB*, and Δ*FosB* (Table 1). Sex effects showed that EtOH-naïve male rats had a higher DLS expression of *Egr1* (*p* = 0.001), *Fos* (*p* = 0.032), *FosB* (*p* < 0.001), and Δ*FosB* (*p* < 0.001) (Figure 6A–D) in comparison to females.

In the VMS, MR1916 exposure resulted in a significant Dose X Sex interaction (*p* = 0.026) and Dose main effect (*p* = 0.042) in *FosB* expression (Figure 7C). Pairwise comparisons showed that MR1916 increased *FosB* expression in male, but not female, rats as compared to their vehicle-injected counterparts (*p* = 0.004). Consequently, MR1916-treated males had a greater *FosB* expression than MR1916-treated females (*p* < 0.001, Figure 7C). Similar trends for MR1916-induced increases in males, but not females, were evident for the other IEGs (Table 1), though the Dose X Sex interactions were not significant. The sex main effects showed that the expression of *Egr1* (*p* = 0.001, Figure 7A), *FosB* (*p* = 0.001, Figure 7C), and Δ*FosB* (*p* = 0.009, Figure 7D) were significantly higher in naïve males than females.

### 3.4. Gene Expression in Post-CIE Rats

#### 3.4.1. MSN Marker Expression

In the DLS, MR1916 exposure dose-dependently increased the expression of *Penk* (*p* = 0.042) in post-CIE rats; pairwise comparisons showed a greater *Penk* expression in rats treated with 0.4 µmol/kg MR1916 vs. vehicle controls (*p* = 0.013; Figure 8D). Sex differences showed that post-CIE female rats had a greater *Penk* expression than male rats (*p* = 0.008, Figure 8D).

There was no significant effect of MR1916 pretreatment on the expression of MSN markers in the VMS in post-CIE male or female rats (Figure 9).

#### 3.4.2. IEG Expression

In the DLS, significant MR1916 X Sex interactions and Dose main effects were seen in the expression of *Fos* (*p* = 0.002 and *p* < 0.001, respectively, Figure 10B) and *FosB* (*p* = 0.022 and *p* = 0.001, respectively, Figure 10C) of post-CIE rats. A pairwise comparison showed that 0.4 µmol/kg MR1916 significantly increased the *Fos* and *FosB* gene expression of female rats in comparison to their vehicle and 0.05 µmol/kg MR1916 conditions (all *p* < 0.001), as well as vs. male rats treated with 0.4 µmol/kg MR1916 (*ps* < 0.001; Figure 10B,C). In contrast, there was no significant effect of MR1916 vs. vehicle control in the expression of *Fos* or *FosB* in post-CIE male rats. Thus, the higher MR1916 dose increased *Fos* and *FosB* expression in post-CIE females (2.19 and 1.83 *z*-score units, respectively), but not males (Table 1). In addition, the Dose main effects (Appendix A) showed that MR1916 also increased the expression of *Egr1* (*p* = 0.001) and Δ*FosB* (*p* = 0.022) in the DLS of post-CIE rats (Figure 10). Pairwise comparisons showed that rats treated with 0.4 µmol/kg MR1916 had a significantly increased *Egr1* expression in the DLS vs. vehicle- (*p* < 0.001) and 0.05 µmol/kg-injected rats (*p* = 0.002, Figure 10A). Sex main effects (Appendix A) reflected that post-CIE females had a higher expression of *Fos* (*p* = 0.029), *FosB* (*p* = 0.002), and Δ*FosB* (*p* = 0.009) compared to male rats in the DLS (Figure 10B–D).

In the VMS, MR1916 administration caused a dose-dependent increase in *Fos* expression (*p* = 0.002); pairwise comparisons showed significant increases in rats treated with 0.4 µmol/kg MR1916 vs. vehicle control (*p* = 0.001) or 0.05 µmol/kg MR1916 (*p* = 0.006, Figure 11B). There were no Sex main effects on IEG expression in the VMS of post-CIE male and female rats.

#### 3.4.3. High- vs. Low-Drinking Post-CIE Male Rats: MSN Marker Expression

A comparison of MR1916 treatment effects in high- vs. low-drinking, post-CIE male rats showed a significant MR1916 X Drinking Level interaction (Appendix A) in the expression of *Tac1* (*p* = 0.001, Figure 12A), *Drd1* (*p* = 0.008, Figure 12B), and *Pde10a* (*p* = 0.041, Figure 12C) in the DLS of post-CIE male rats. A pairwise comparison showed that the higher MR1916 dose (0.4 µmol/kg) decreased *Drd1* in low-drinking post-CIE males vs. both their vehicle control (*p* = 0.045) and lower-dose-treated counterparts (*p* = 0.011). In contrast, high-drinking males injected with the higher MR1916 dose had a significantly higher *Drd1* expression than those injected with the lower dose (*p* = 0.043) and, consequently, showed a higher *Drd1* expression than low-drinking males injected with the higher dose (*p* = 0.001, Figure 12B). Similarly, low drinkers treated with the higher MR1916 dose showed a significantly lower (*p* < 0.05) *Pde10a* expression than high drinkers that received the same dose (Figure 12). A Drinking Level main effect (Appendix A) reflected that high-drinking post-CIE male rats had a significantly higher *Drd1* expression in the DLS than low-drinking males (*p* = 0.026, Figure 12B). An inspection of Figure 12 descriptively showed that these divergent effects were largest in high-dose-MR1916-treated rats. Accordingly, there was a significant direct correlation between the baseline drinking level with DLS *Drd1* (*r*[7] = 0.85, *p* = 0.016) and *Pde10a* expression (*r*[7] = 0.87, *p* = 0.011) after MR1916 high-dose treatment. These relations were not seen under vehicle or low-dose treatment (Appendix A).

There was no significant effect of MR1916 in the expression of MSN markers in the VMS (Figure 13).

Although no female rat was defined as Low-Drinking by our definition, a correlation analysis showed that higher baseline drinking was associated with a significantly higher *Drd2* expression (*r*[7] = 0.77, *p* = 0.041) in the DLS, but not VMS (*r*[7] = 0.06, *p* = 0.89) in vehicle-treated females. Furthermore, in females, baseline drinking was inversely correlated with the expression of *Pde10a*, *Drd1*, and *Penk* in the VMS after high-dose MR1916 treatment (*r*s = −0.81, −0.81, and −0.88, respectively; *p*s = 0.028, 0.027, and 0.009, Appendix A).

#### 3.4.4. High- vs. Low-Drinking Post-CIE Male Rats: IEG Expression

Regarding IEG expression differences between high- and low-drinking post-CIE male rats, a Drinking Level main effect (Appendix A) showed that high drinkers had a higher *FosB* expression in the DLS than low-drinking rats (*p* = 0.021). An inspection of Figure 14C descriptively showed this effect was largest in high-dose-MR1916-treated rats. Accordingly, there was a significant direct correlation of the baseline drinking level with DLS *FosB* expression after MR1916 high-dose treatment (*r*[7] = 0.82, *p* = 0.025), but not vehicle (*r* = −0.01) or low-dose (*r* = 0.17) treatment (Appendix A).

In the VMS, there was a significant MR1916 X Drinking Level interaction (Appendix A) in the expression of *Egr1* (*p* = 0.003, Figure 15A) and *Fos* (*p* = 0.016, Figure 15B) and a main effect of Drinking Level (Appendix A) on *Egr1* expression (*p* = 0.031, Figure 15A). Irrespective of drinking level, polynomial contrasts showed that MR1916 treatment altered the *Egr1* expression per a quadratic dose–response (*p* < 0.001, Figure 15A), and the *Fos* expression in a linearly increasing, dose-dependent manner (*p* = 0.028, Figure 15B). Pairwise comparisons showed that the higher MR1916 dose increased *Egr1* expression in low-drinking males vs. their respective vehicle (*p* < 0.001) and low-dose counterparts (*p* < 0.001). Unlike in low-drinking males, the high MR1916 dose did not alter the *Egr1* expression of high-drinking males vs. vehicle. Rather, low-dose MR1916 decreased the expression of *Egr1* by high drinkers vs. their respective vehicle control (*p* = 0.035). Unlike in the DLS, there were no significant Pearson correlations of the baseline intake to IEG expression after MR1916 treatment in males.

Under vehicle conditions, high-drinking males had a greater VMS *Egr1* expression than low drinkers (*p* = 0.001, Figure 15A). Similarly, high-drinking post-CIE males had a higher VMS expression of Δ*FosB* than low drinkers (Drinking Level main effect, *p* = 0.023, Figure 15D). These relations were significant in the subgroup analyses by the high- vs. low-drinking category, but not the continuous correlation analyses (Appendix A).

Although no female rat was defined as Low-Drinking, the correlation analysis showed that higher baseline drinking was associated with significantly higher *Egr1* expression (*r*[7] = 0.79, *p* = 0.036) in the DLS, but not VMS (*r*[7] = 0.27, *p* = 0.52), of vehicle-treated females. Furthermore, in females, baseline drinking was inversely correlated with the expression of all IEGs in the VMS after high-dose MR1916 treatment (*r*s = −0.89, −0.89, −0.91, and −0.83 for *Egr1*, *Fos*, *FosB*. and Δ*FosB*, respectively; *p*s = 0.008, 0.008, 0.004, and 0.02, respectively, Appendix A).

## 4. Discussion

We have found that the systemic administration of the drug-like PDE10A inhibitor MR1916 dose-relatedly decreased EtOH self-administration in high-drinking, but not low-drinking, post-CIE male rats. In both EtOH-naïve and post-CIE rats, MR1916 dose-relatedly increased striatal neuronal activation, evident as the increased expression of the IEGs *Egr1*, *Fos*, *FosB*, and Δ*FosB* in the DLS, where PDE10A is most abundant, as well as of Fos family IEGs (*Fos* and *FosB*) within the VMS. MR1916 also increased the expression of the MSN markers *Pde10a*, *Drd1*, and *Drd2* in the DLS, but not VMS, of naïve rats, supporting the hypothesis that it activates both direct and indirect pathway MSNs. Interestingly, MR1916 did not induce the same MSN markers across post-CIE rats. Instead, a dose sufficient to reduce EtOH self-administration increased the DLS expression of *Penk*, an iMSN subpopulation marker. It also modulated pan-MSN (*Pde10a*) and dMSN (*Drd1*, *Tac1*) markers in relation to drinking level, decreasing them for low drinkers while tending to increase them for high drinkers. Finally, sex differences were seen in the behavioral and molecular actions of the PDE10A inhibitor. Whereas it reduced self-administration in high-drinking males, MR1916 elicited a quadratic Dose contrast effect on both EtOH intake and preference in females, reflecting that the low 0.5 µmol/kg dose marginally increased them in low drinkers. MR1916 also increased the DLS expression of IEGs more in male than female EtOH-naïve rats but had the opposite pattern in post-CIE rats. Altogether, the results indicate that MR1916 can reduce EtOH self-administration in high-drinking males at doses that promote striatal MSN activation and suggest that its actions may differ in relation to dose, sex, EtOH drinking level, and EtOH history.

MR1916 reduced EtOH self-administration in post-CIE males in a log-linear dose-related fashion (Figure 2A). This effect reflected MR1916’s suppressive action in high-drinking males; in low-drinking males, the compound instead exerted a significant quadratic Dose contrast on self-administration (Figure 2A,B), yielding a Dose X Drinking Level interaction. The 0.6 g/kg threshold of EtOH intake that we used to define “high-drinking” has been used previously by other groups [16,47] and reflects that EtOH doses of this magnitude are needed to promote place conditioning and pharmacologically relevant blood EtOH concentrations in rats. Effects were unlikely to be due to non-specific locomotor suppression because we performed a catalepsy bar test to avoid testing doses that generally impaired movement initiation. This bar test is sensitive to cataleptic-like actions of other PDE10A inhibitors and functionally related D2 receptor antagonists [19] and, here, similarly identified mild cataleptic-like activity of the (excluded) 0.8 µmol/kg dose of MR1916. At the tested doses, MR1916 also did not significantly affect concurrent water self-administration, supporting the interpretation that the reduced EtOH self-administration was not due to general motor suppression. The current results during post-dependence extend upon our previous findings with the PDE10A inhibitor TP-10 in acute withdrawal, genetically selected alcohol preference, and post-traumatic stress models of high EtOH administration [8].

The present findings also align with prior work that link chronic EtOH, blunted cAMP signaling cascades, and alcohol drinking phenotypes. Thus, chronic EtOH blunts induction of several adenyl cyclase isoforms [52] and increases striatal PDE10A enzymatic availability, which would curtail cAMP action [15]. Mouse models deficient in adenyl cyclase (AC7) [53], protein kinase A [54], or cAMP-response element binding protein (CREB) effector action [55] along the cAMP cascade reportedly show increased EtOH intake or preference. Humans with AUD show impaired induction of adenyl cyclase 7 as well as gene variants in *ADCY7* and *PDE4B*, a phosphodiesterase that, like PDE10A, hydrolyzes cAMP [9,56,57]. Similar to the present work, PDE4 inhibitors, including rolipram, Ro 20-1724, roflumilast, D159687, and apremilast, reduced EtOH intake in rodents or people [58,59,60]. Likewise, ibudilast, a pan-PDE inhibitor, reduced EtOH consumption in EtOH-dependent rodents as well as craving and withdrawal symptoms in human patients with AUD [61,62]. PDE10A represents an intriguing molecular target for modulating striatal cAMP signaling during post-dependence because of the highly differential expression of PDE10A in MSNs. Given this localization, PDE10A inhibitors might yield fewer “off-system” effects than inhibitors of more widely expressed PDEs, such as PDE4.

MR1916 produced differential effects in male versus female rats, in accordance with previous in vitro findings with other PDE10A inhibitors [42]. Whereas the compound dose-dependently reduced EtOH self-administration in high-drinking males, females showed a quadratic Dose contrast independent of drinking level (Figure 2A,B). MR1916 also induced a greater DLS expression of IEGs in EtOH-naïve males than females (*z*-score increase vs. vehicle across IEGs of [*M*+SEM] 1.26 + 0.10 vs. 0.31 + 0.11, *p* = 0.0005), a dimorphism opposite to that seen in post-CIE rats (0.52+0.06 vs. 1.28+0.20, *p* = 0.01; Table 1). Perhaps underlying the sex differences in PDE10A inhibitor action, EtOH-naïve, but not post-CIE, males showed a greater striatal *Pde10a* expression than females in both the DLS and VMS. The sex difference in MR1916 action also might reflect known sex differences in PDE10A protein synaptic expression, since previous studies have shown that PDE10A expression is higher in the DLS in female rats with a history of alcohol consumption, in comparison to male rats [63]. Consequently, higher doses of MR1916 might be needed to inhibit PDE10A function in female rats. Furthermore, *PDE10A* has been identified as a candidate gene contributing to strong sex-specific differences in thyroid function and related pathologies [64], reiterating a potential sexual dimorphism of PDE10A function. Sex differences have also been observed in some studies with apremilast, a PDE4 inhibitor [65]. A related consideration is that sex differences in PDE inhibitor action generally might be due to sex differences in MSN activity and associated cAMP/cGMP cascades, independent of the PDEs per se. Thus, MSNs recorded from prepubertal females show increased excitation compared with those from males in the caudate–putamen and NAc core [66].

At the studied doses, MR1916 induced a greater IEG expression in the DLS than VMS of both naïve and post-CIE rats. Where significant increases were seen, IEG expression increased by 1 z-score in the DLS vs. 0.7 for the VMS; in addition, more DLS IEG increases were statistically significant. Like the present findings, other PDE10A inhibitors elicited a greater induction of *Fos* in the lateral compared to the medial striatum [20,21] and greater *Egr1* in DLS than VMS [19]. These regional differences may reflect the greater expression [9,67,68] (and possibly functional role) of PDE10A in humans and rodents within the DLS as compared to the VMS. Under conditions where significant IEG induction was observed, Δ*FosB* showed the smallest responses to acute MR1916 (z-score increase: 0.88 + 0.08 vs. 1.20 + 0.14 for the other IEGs), perhaps reflecting its greater utility as an indicator of longer-term activation [69]. *Fos* and *Egr1* showed the largest induction (mean *z*-score increases of 1.37 and 1.14, respectively), consistent with reports that the striatal expression of these IEGs is altered by PDE10A inhibition or genetic deletion [19,21,70]. Consistent with PDE10A’s mechanism, these genes are transcriptionally regulated by cAMP and cGMP cascade elements [23,24,25,26,27,28,29,30,31,32,33].

The MSN markers most induced by MR1916 in EtOH-naïve rats were *Pde10a* (pan-MSN marker), *Drd1* (dMSNs), and *Drd2* (iMSNs), all differentially in the DLS over the VMS. Their similar degree of induction suggests that, at these doses, MR1916 elicited balanced MSN facilitation, as opposed to iMSN- or dMSN-biased influence [41]. On the other hand, in post-CIE rats, the dose of MR1916 that reduced EtOH self-administration instead differentially induced *Penk*, an iMSN subpopulation marker (Figure 8D). The present results raise the hypothesis that alcohol history or intake levels may alter the PDE10A action on MSN subpopulations.

Relatedly, MR1916-induced effects in post-CIE rats on pan-MSN (*Pde10a*) and dMSN (*Drd1* and *Tac1*) markers diverged significantly in relation to drinking level; they decreased for low drinkers but not for high drinkers. Whether MR1916’s differential action to reduce dMSN markers in low-drinking male rats relates to its inability to reduce their EtOH self-administration is unclear. But, overall, in post-CIE rats and, especially, low drinkers, MR1916 elicited more iMSN-biased facilitation, similar to inhibitors with fast enzyme off-rate kinetics like TAK-063 [41]. Fast off-rate PDE10A inhibitors have been proposed to have less relative effect on the dMSN pathway (more iMSN-biased) than slow off-rate inhibitors because dMSNs may have greater constitutive cAMP/cGMP or PDE10A activity. Thus, more sustained PDE10A inhibition may be required to influence them [41]. Accordingly, understanding alcohol-associated differences or adaptations in the activation and phosphodiesterase biology of MSN subpopulations may clarify the PDE10A-related pathophysiology or treatment of AUD.

In this context, the different expression of some MSN and IEG markers in relation to drinking level are of interest. High-drinking male rats, as compared to low drinkers, showed a significantly increased DLS *Drd1* expression, as well as reliably increased VMS *Egr1* and Δ*FosB*, with similar trends evident for *Fos* and *FosB*. Broadly consistent with our IEG activation marker findings, previous studies found higher levels of *Fos* in the VMS of EtOH-preferring rats [71] and of rats with a higher mean EtOH intake [72]. *Fos* expression was also greater in the VMS of male rats selectively bred for high EtOH consumption vs. low-consuming rats [73]. While our findings resemble the reviewed studies, we cannot infer causality based on our current data. We also cannot conclude whether the differences in VMS activation between high and low drinkers resulted from their different drinking history or, alternatively, was a pre-existing correlate or determinant of their subsequent differences in EtOH self-administration upon access. Furthermore, due to the reduced sample sizes in the subgroup analysis, we cannot rule out that small effects may have been missed, or mean differences produced by a few outlying scores. This individual variability in drug response may be useful to study further using a correlation analysis of biological and behavioral responses.

In summary, non-cataleptic doses of MR1916 reduced the EtOH self-administration of high-drinking, post-CIE males in a linear fashion but showed a quadratic relation in females and low-drinking males, tending to increase it at the low 0.05 µmol/kg dose. MR1916 increased striatal neuronal activation, most evident for the IEG markers *Egr1*, *Fos*, and *FosB* and in the DLS where PDE10A is most abundant. Alcohol and sex may modulate molecular effects of PDE10A inhibition because the greatest IEG induction was seen in ethanol-naïve males and post-CIE females. MR1916 also modulated striatal MSN markers differentially in relation to alcohol factors, eliciting the “balanced” induction of dMSN and iMSNs (*Pde10A*, *Drd1*, and *Drd2*) in ethanol-naïve rats but iMSN subpopulation induction (*Penk*) in post-CIE ones and even a reduced dMSN marker expression (*Drd1* and *Tac1*) in post-CIE low drinkers. Finally, high-drinking, post-CIE male rats, as compared to low drinkers, had a higher DLS *Drd1*, and VMS IEG expression, especially *Egr1* and Δ*FosB.* A key future direction will be to determine if PDE10A inhibitors like MR1916 normalize the effects of acute or chronic EtOH self-administration on MSN marker and IEG expression, as well as the EtOH drinking effects on cAMP and cGMP cascades themselves. Altogether, our study in a post-CIE model with construct validity for post-dependent AUD treatment has identified sex- and alcohol-related actions of a drug-like PDE10A inhibitor to alter ethanol self-administration, as well as IEG activation and MSN marker expression in DLS and VMS circuits thought to subserve ethanol reinforcement and compulsive drinking. Combined with previous studies of PDE10A and PDE4 inhibitors in excessive drinking models, the results support the hypothesis that phosphodiesterase-targeted treatments that modulate MSN signaling may have therapeutic potential for AUD.

## Figures and Tables

**Figure 1 cells-13-00321-f001:**
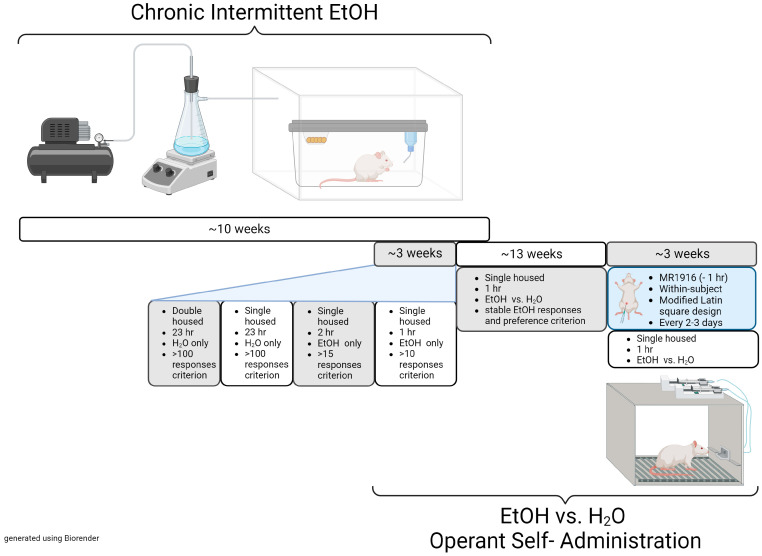
Experimental timeline of the chronic intermittent ethanol exposure and operant self-administration training behavioral model. CIE phase = 10–14 weeks; training phase = 2–3 weeks; EtOH responses and preference stabilization phase = 2–18 weeks; MR1916 treatment phase: 3–4 weeks.

**Figure 2 cells-13-00321-f002:**
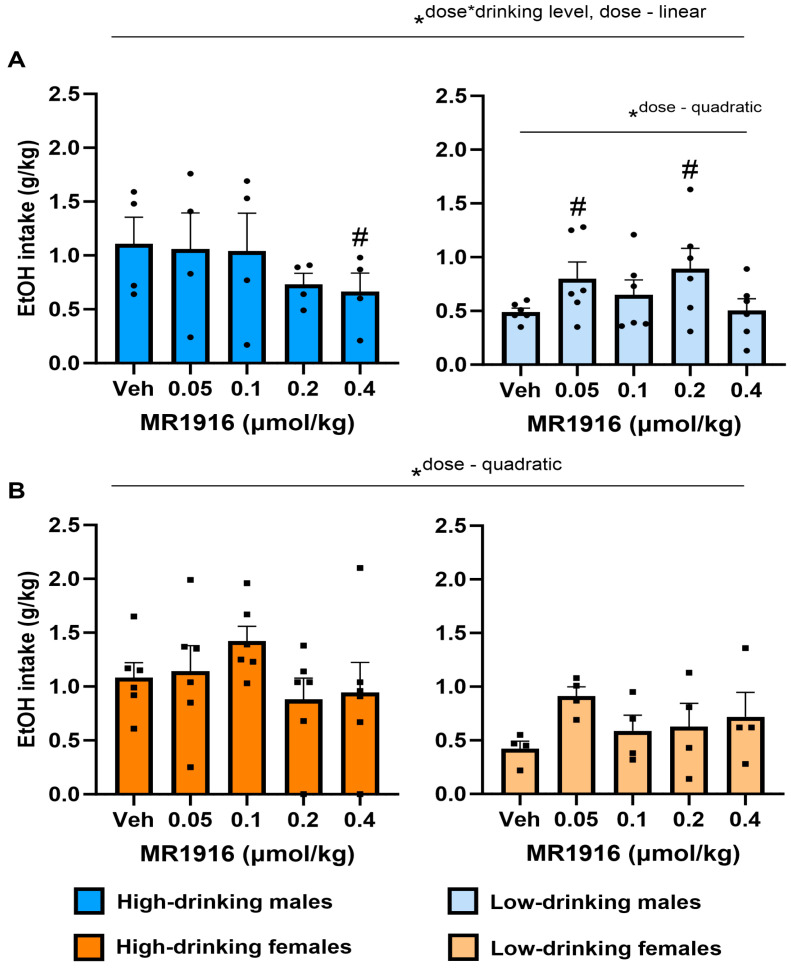
Effects of MR1916 pretreatment (−60 min) on total EtOH intake in post-CIE (**A**) male and (**B**) female rats during 2-lever EtOH vs. water self-administration. Left panels show high-drinking (>0.6 g/kg at baseline) rats; right panels show low-drinking rats. Histograms show group *M*+SEM, and scatter (dot symbols) shows the scores of individual subjects. *N* = 10/sex; *ANOVA within-subject contrast. Doses marked by the pound sign differ significantly from the respective vehicle control. #, * *p* < 0.05.

**Figure 3 cells-13-00321-f003:**
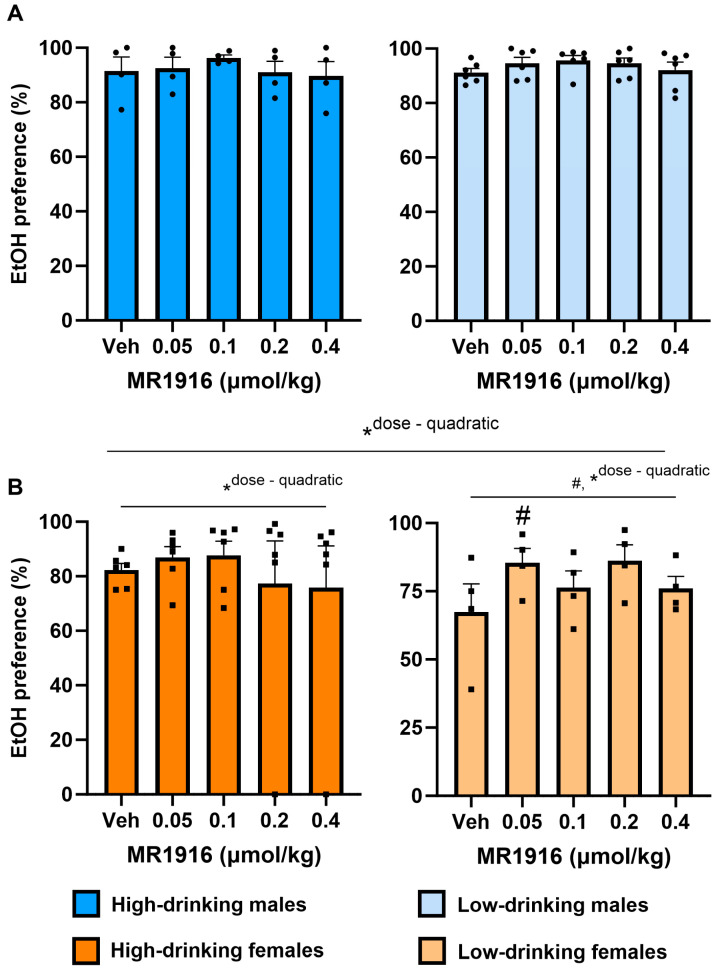
Effects of MR1916 pretreatment (−60 min) on EtOH reinforcer preference in post-CIE (**A**) male and (**B**) female rats during 2-lever EtOH vs. water self-administration. Left panels show high-drinking rats (>0.6 g/kg at post-CIE baseline) rats; right panels show low-drinking rats. Histograms show group *M*+SEM, and scatter (dot symbols) shows the scores of individual subjects. *N* = 10/sex; # ANOVA within-subject effect. * ANOVA within-subject contrast. Doses marked by the pound sign differ significantly from the respective vehicle control. #, * *p* < 0.05.

**Figure 4 cells-13-00321-f004:**
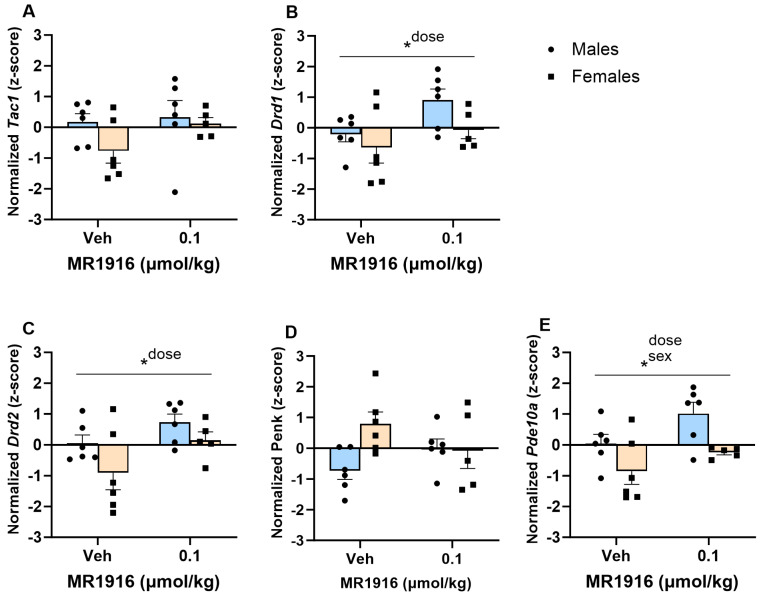
Effects of MR1916 on gene expression of (**A**) substance P (*Tac1*), (**B**) dopamine receptor 1 (*Drd1*), (**C**) dopamine receptor 2 (*Drd2*), (**D**) enkephalin (*Penk*), and (**E**) phosphodiesterase 10a (*Pde10a*), in the dorsal–lateral striatum of EtOH-naïve male (blue fill) and female (orange) rats 90 min after treatment. Each bar represents the mean *z*-score ± standard error, showing individual values. *N* = 5–6/treatment/sex. Graph marked by an asterisk and a line have the indicated significant ANOVA effects. * *p* < 0.05.

**Figure 5 cells-13-00321-f005:**
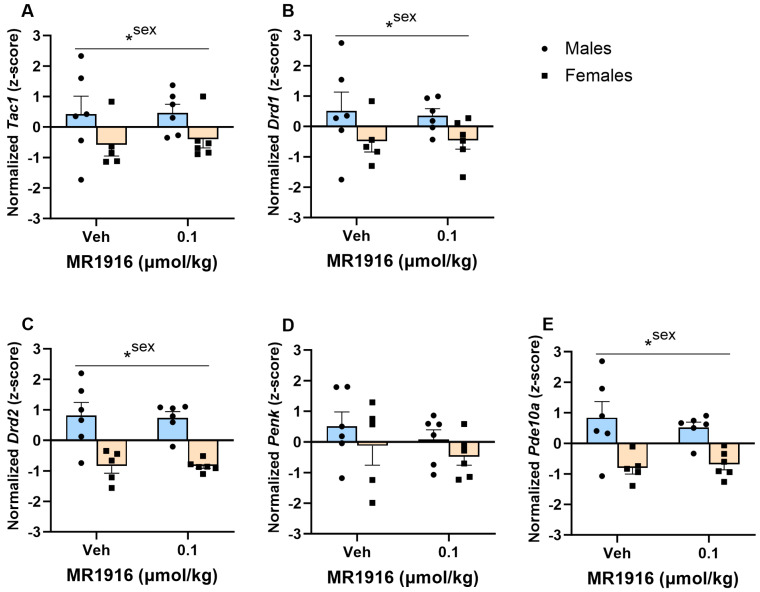
Effects of MR1916 on gene expression of (**A**) substance P (*Tac1*), (**B**) dopamine receptor 1 (*Drd1*), (**C**) dopamine receptor 2 (*Drd2*), (**D**) enkephalin (*Penk*), and (**E**) phosphodiesterase 10a (*Pde10a*), in the ventromedial stratum of EtOH-naïve male (blue fill) and female (orange) rats 90 min after treatment. Each bar represents the mean *z*-score ± standard error, showing individual values. *N* = 5–6/treatment/sex. Graph marked by an asterisk and a line have the indicated significant ANOVA effects. * *p* < 0.05.

**Figure 6 cells-13-00321-f006:**
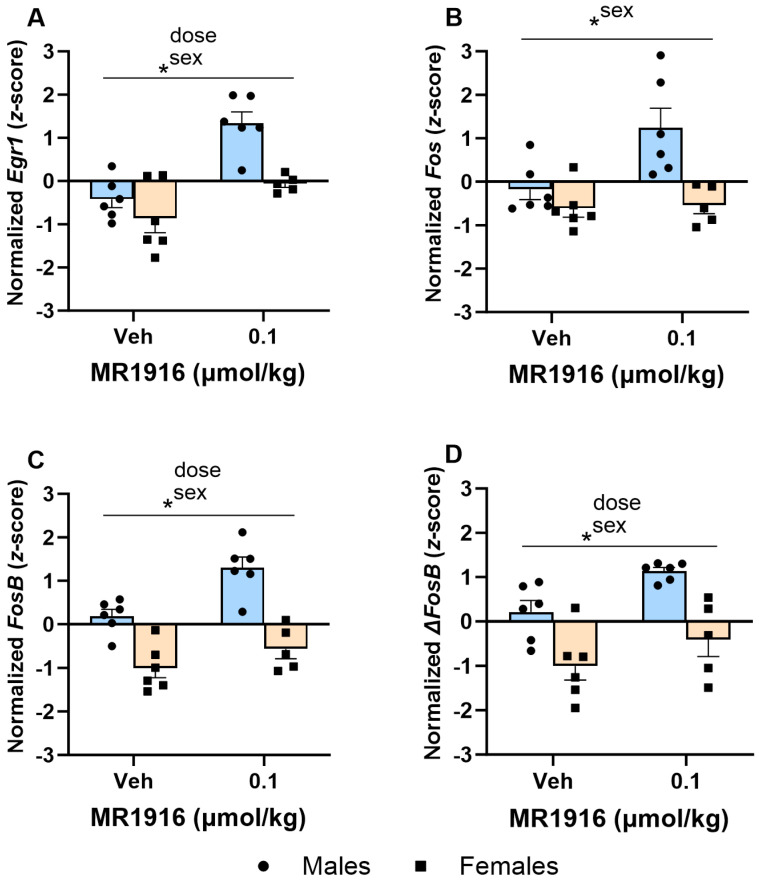
Effects of MR1916 on gene expression of (**A**) early growth response protein 1 (*Egr1*), (**B**) FBJ murine osteosarcoma viral oncogene homolog (*Fos*), (**C**) FBJ murine osteosarcoma viral oncogene homolog B (*FosB*), and (**D**) delta FBJ murine osteosarcoma viral oncogene homolog B (Δ*FosB*), in the dorsal–lateral striatum of EtOH-naïve male (blue fill) and female (orange) rats 90 min after treatment. Each bar represents the mean *z*-score ± standard error, showing individual values. *N* = 5–6/treatment/sex. Graph marked by an asterisk symbol and a line have the indicated significant ANOVA effects. * *p* < 0.05.

**Figure 7 cells-13-00321-f007:**
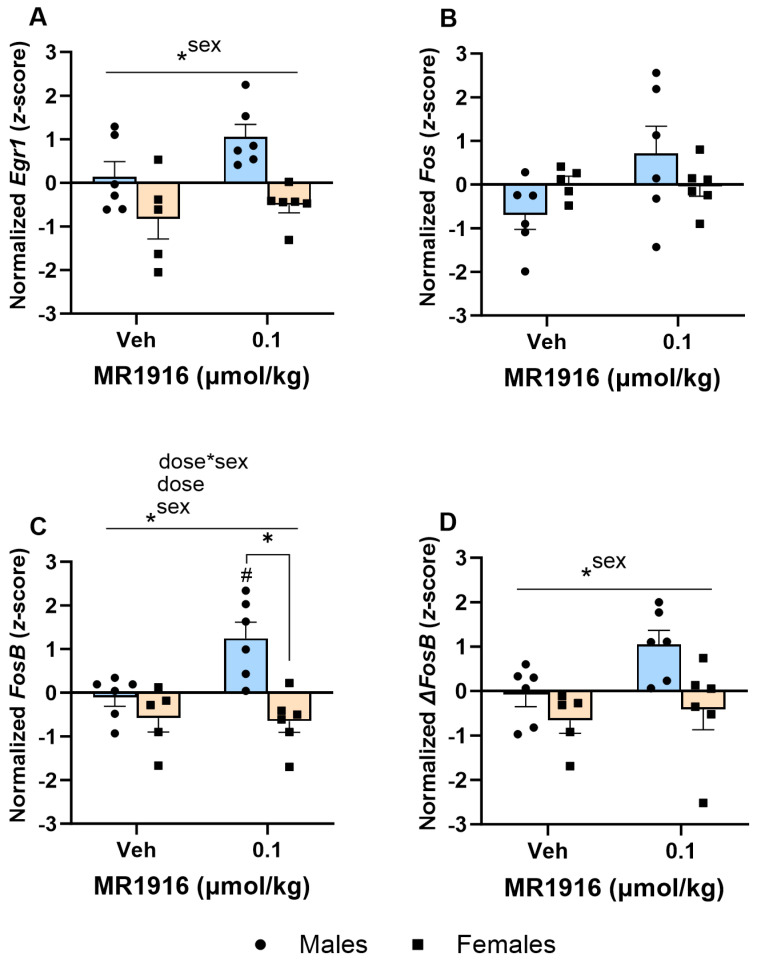
Effects of MR1916 on gene expression of (**A**) early growth response protein 1 (*Egr1*), (**B**) FBJ murine osteosarcoma viral oncogene homolog (*Fos*), (**C**) FBJ murine osteosarcoma viral oncogene homolog B (*FosB*), and (**D**) delta FBJ murine osteosarcoma viral oncogene homolog B (Δ*FosB*), in the ventromedial stratum of EtOH-naïve male (blue fill) and female (orange) rats after 90 min of treatment. Each bar represents the mean *z*-score ± standard error, showing individual values. *N* = 5–6/treatment/sex. Graph marked by the asterisk sign and a line have a significant between-subject effect (ANOVA). Bars marked by the asterisk sign and a bracket are significantly different from each other (LSD post hoc). Bars marked by the pound sign are significantly different from the respective vehicle control (LSD post hoc). #, * *p* < 0.05.

**Figure 8 cells-13-00321-f008:**
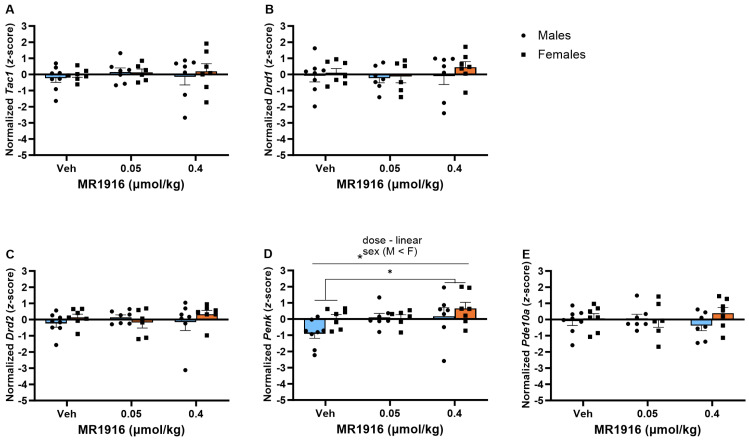
Effects of MR1916 on gene expression of (**A**) substance P (*Tac1*), (**B**) dopamine receptor 1 (*Drd1*), (**C**) dopamine receptor 2 (*Drd2*), (**D**) enkephalin (*Penk*), and (**E**) phosphodiesterase 10a (*Pde10a*), in the dorsal–lateral striatum of post-CIE male (blue fill) and female (orange) rats 90 min after treatment. Each bar represents the mean *z*-score ± standard error, showing individual values. *N* = 6–8/treatment/sex. Graph marked by the star sign and a line have a significant between-subject effect (ANOVA). Groups (males + females) marked by the star sign and a bracket are significantly different from each other (LSD post hoc). * *p* < 0.05.

**Figure 9 cells-13-00321-f009:**
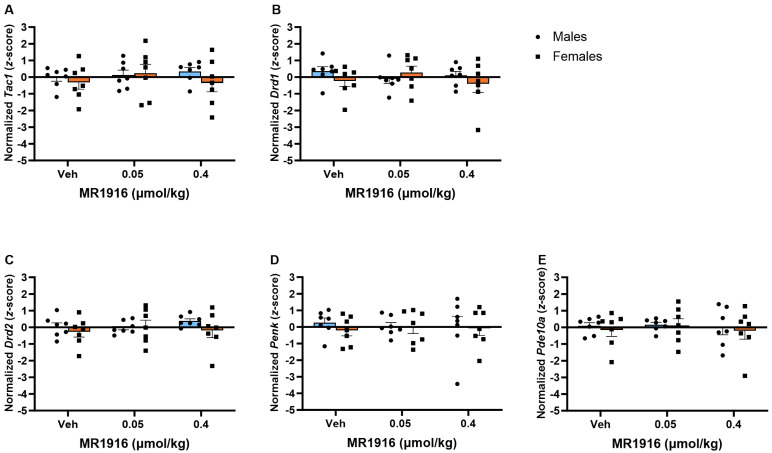
Effects of MR1916 on gene expression (**A**) substance P (*Tac1*), (**B**) dopamine receptor 1 (*Drd1*), (**C**) dopamine receptor 2 (*Drd2*), (**D**) enkephalin (*Penk*), and (**E**) phosphodiesterase 10a (*Pde10a*), in the ventromedial striatum of EtOH post-CIE male (blue fill) and female (orange) rats 90 min after treatment. Each bar represents the mean *z*-score ± standard error, showing individual values. *N* = 6–8/treatment/sex.

**Figure 10 cells-13-00321-f010:**
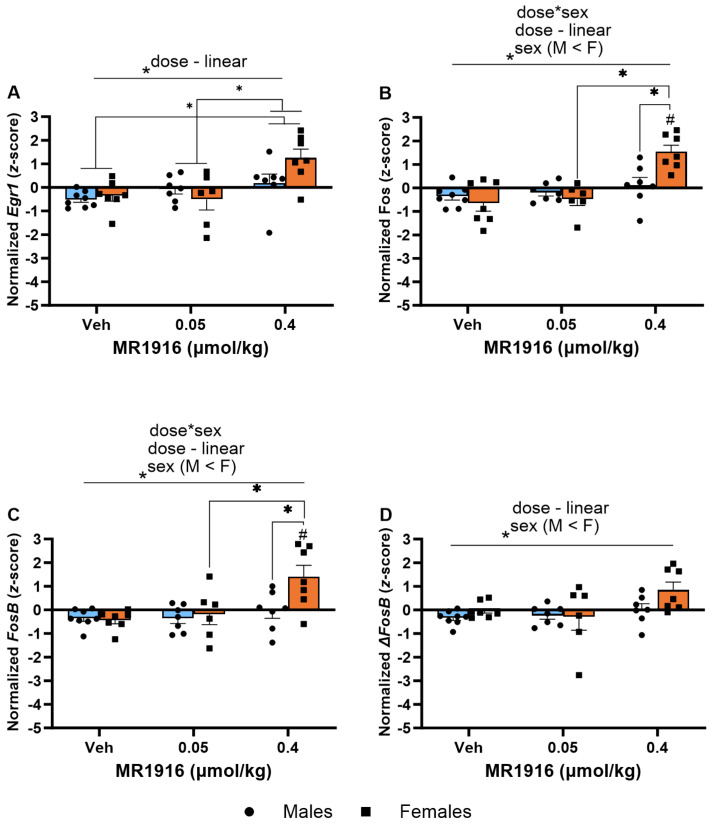
Effects of MR1916 on gene expression of (**A**) early growth response protein 1 (*Egr1*), (**B**) FBJ murine osteosarcoma viral oncogene homolog (*Fos*), (**C**) FBJ murine osteosarcoma viral oncogene homolog B (*FosB*), and (**D**) delta FBJ murine osteosarcoma viral oncogene homolog B (Δ*FosB*), in the dorsal–lateral striatum of post-CIE male (blue fill) and female (orange) rats 90 min after treatment. Each bar represents the mean *z*-score ± standard error, showing individual values. *N* = 6–8/treatment/sex. Graph marked by the asterisk and a line have the indicated significant ANOVA effects. Asterisks with brackets also indicate significant pairwise differences between indicated groups (LSD test). Bars marked by the pound sign differ from their respective vehicle control (LSD post hoc). #, * *p* < 0.05.

**Figure 11 cells-13-00321-f011:**
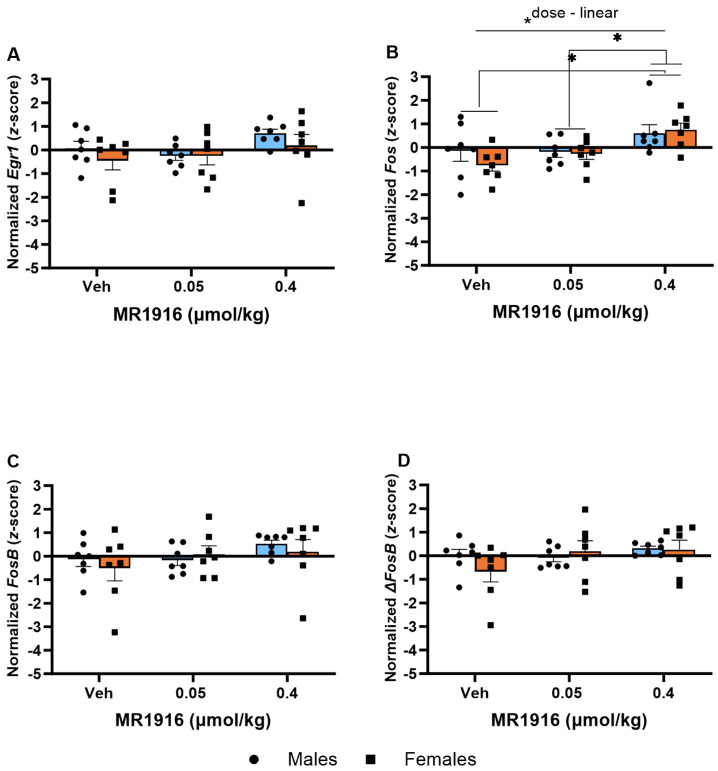
Effects of MR1916 on gene expression of (**A**) early growth response protein 1 (*Egr1*), (**B**) FBJ murine osteosarcoma viral oncogene homolog (*Fos*), (**C**) FBJ murine osteosarcoma viral oncogene homolog B (*FosB*), and (**D**) delta FBJ murine osteosarcoma viral oncogene homolog B (Δ*FosB*), in the ventromedial striatum of post-CIE male (blue fill) and female (orange) rats 90 min after treatment. Each bar represents the mean *z*-score ± standard error, showing individual values. *N* = 6–8/treatment/sex. Graph marked by the asterisk and a line have the indicated significant ANOVA effects. Asterisks with brackets also indicate significant pairwise differences between indicated groups (LSD test). * *p* < 0.05.

**Figure 12 cells-13-00321-f012:**
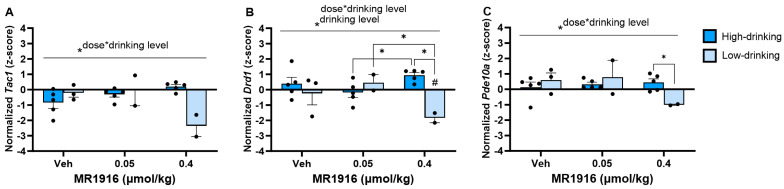
Effects of MR1916 on gene expression of (**A**) substance P (*Tac1*), (**B**) dopamine receptor 1 (*Drd1*), and (**C**) phosphodiesterase 10a (*Pde10a*), in the dorsal–lateral striatum of post-CIE male rats 90 min after treatment. Each bar represents the mean *z*-score ± standard error, showing individual values. *N* = 2–5/treatment/drinking level. Graph marked by the asterisk and a line have the indicated significant ANOVA effects. Asterisks with brackets also indicate significant pairwise differences between indicated groups (LSD or Dunnet’s T3 test). Bars marked by the pound sign are significantly different from their respective vehicle control (LSD test). #, * *p* < 0.05.

**Figure 13 cells-13-00321-f013:**
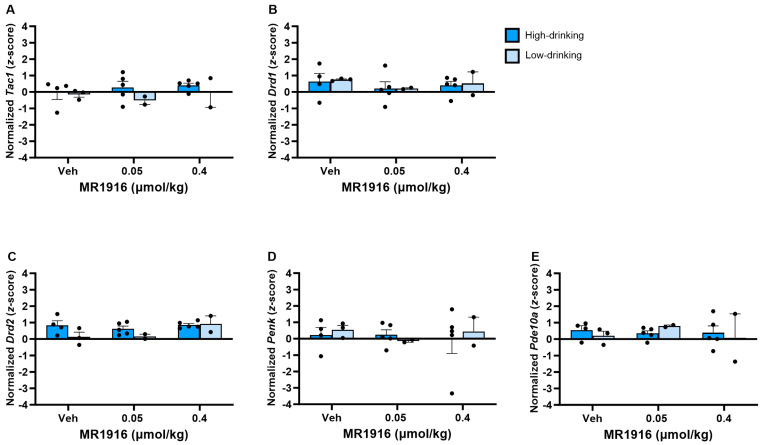
Effects of MR1916 on gene expression of (**A**) substance P (*Tac1*), (**B**) dopamine receptor 1 (*Drd1*), (**C**) dopamine receptor 2 (*Drd2*), (**D**) enkephalin (*Penk*), and (**E**) phosphodiesterase 10a (*Pde10a*), in the ventromedial striatum of post-CIE male rats 90 min after treatment. Each bar represents the mean *z*-score ± standard error, showing individual values. *N* = 2–5/treatment/drinking level.

**Figure 14 cells-13-00321-f014:**
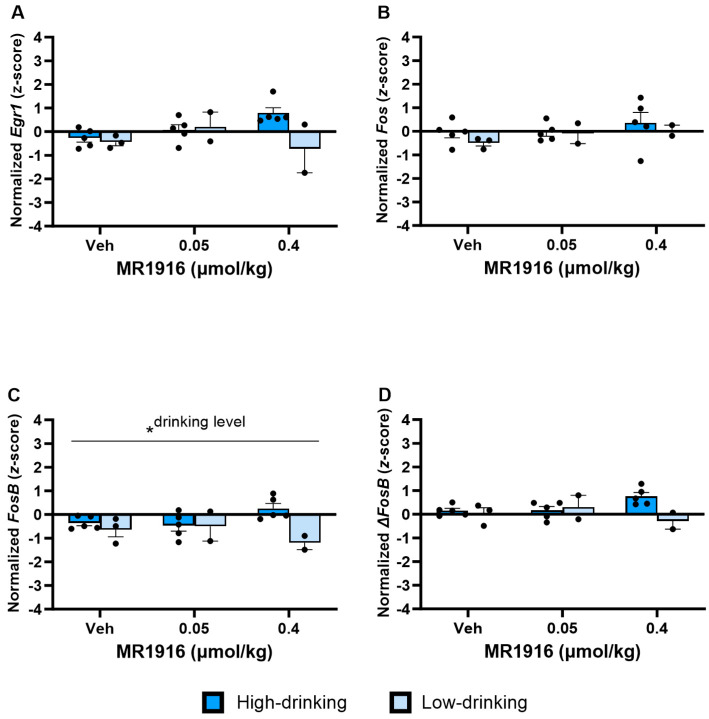
Effects of MR1916 on gene expression of (**A**) early growth response protein 1 (*Egr1*), (**B**) FBJ murine osteosarcoma viral oncogene homolog (*Fos*), (**C**) FBJ murine osteosarcoma viral oncogene homolog B (*FosB*), and (**D**) delta FBJ murine osteosarcoma viral oncogene homolog B (Δ*FosB*), in the dorsal–lateral striatum of post-CIE male rats 90 min after treatment. Each bar represents the mean *z*-score ± standard error, showing individual values. *N* = 2–5/treatment/drinking level. Graph marked by the asterisk and a line have the significant indicated ANOVA effect. * *p* < 0.05.

**Figure 15 cells-13-00321-f015:**
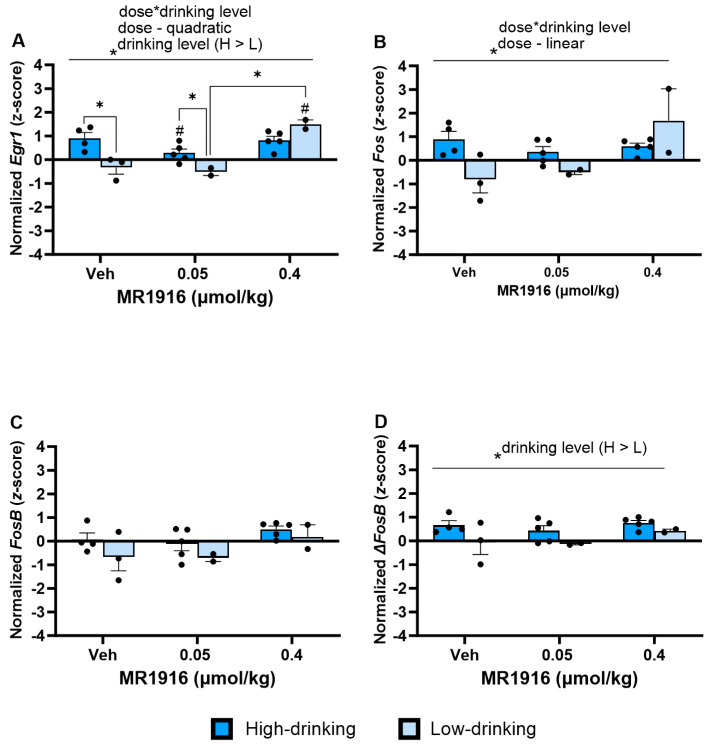
Effects of MR1916 on gene expression of (**A**) early growth response protein 1 (*Egr1*), (**B**) FBJ murine osteosarcoma viral oncogene homolog (*Fos*), (**C**) FBJ murine osteosarcoma viral oncogene homolog B (*FosB*), and (**D**) delta FBJ murine osteosarcoma viral oncogene homolog B (Δ*FosB*), in the ventromedial striatum of post-CIE male rats after 90 min of treatment. Each bar represents the mean *z*-score ± standard error, showing individual values. *N* = 2–5/treatment/drinking level. Graphs marked by the asterisk and a line have the significant indicated ANOVA effect. Bars marked by the asterisk and a bracket differ significantly from each other (LSD test). Bars marked by the pound sign differ significantly from their respective vehicle control (LSD test) #, * *p* < 0.05.

**Table 1 cells-13-00321-t001:** Immediate-early gene expression *z*-score differences (treated—vehicle) and *p*-values in the dorsolateral (DLS) and ventromedial (VMS) striatum of naïve and post-CIE male and female rats. For natural log-transformed data, *z*-score differences were back-transformed.

		Naïve						
		**DLS**	**Males**	***p*-value**	**Females**	***p*-value**		
		*Egr1*	1.76	<0.001	0.80	0.037		
		*Fos* (ln)	1.32	0.003	0.28	0.876		
		*FosB*	1.12	0.001	0.45	0.164		
		Δ*FosB*	0.92	0.026	0.60	0.149		
		**VMS**	**Males**	***p*-value**	**Females**	***p*-value**		
		*Egr1*	0.91	0.054	0.33	0.494		
		*Fos* (ln)	1.54	0.019	−0.12	0.899		
		*FosB*	1.35	0.004	−0.07	0.87		
		Δ*FosB*	1.13	0.03	0.25	0.63		
	**Post-CIE**							
	**Males**				**Females**			
**DLS**	**0.05 µmol/kg**	***p*-value**	**0.4 µmol/kg**	***p*-value**	**0.05 µmol/kg**	***p*-value**	**0.4 µmol/kg**	***p*-value**
*Egr1*	0.44	0.292	0.69	0.103	−0.19	0.677	1.60	0.001
*Fos*	0.16	0.64	0.48	0.176	0.14	0.712	2.19	<0.001
*FosB*	−0.01	0.973	0.30	0.445	0.21	0.624	1.83	<0.001
Δ*FosB*	0.10	>0.05	0.36	>0.05	−0.32	>0.05	0.85	>0.05
	**Males**				**Females**			
**VMS**	**0.05 µmol/kg**	***p*-value**	**0.4 µmol/kg**	***p*-value**	**0.05 µmol/kg**	***p*-value**	**0.4 µmol/kg**	***p*-value**
*Egr1*	−0.32	0.496	0.63	0.187	0.20	0.674	0.64	0.185
*Fos*	−0.05	0.913	0.74	0.104	0.47	0.293	1.50	0.002
*FosB*	−0.03	0.955	0.65	0.238	0.59	0.287	0.69	0.213
Δ*FosB*	−0.08	867	0.31	0.519	0.85	0.083	0.93	0.059

## Data Availability

We have provided all test data reported here at the OSF website (https://osf.io/p64mr/files/osfstorage/65493b4d253a7403e6a0ed8d) (provided 6 November 2023). Additional data and statistical output are available upon request from the corresponding author.

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
