# Peer review of "Effects of the Phosphodiesterase 10A Inhibitor MR1916 on Alcohol Self-Administration and Striatal Gene Expression in Post-Chronic Intermittent Ethanol-Exposed Rats"

_cells, 2024, doi:10.3390/cells13040321_

Round 1
Reviewer 1 Report
Comments and Suggestions for Authors
In this manuscript, Bertotto and colleagues uncover a potentially important role for phosphodiesterase 10A (PDE10A), likely in the striatum, in post-dependent ethanol drinking, and specifically. In a series of elegant experiments, they found that the phosphodiesterase 10A (PDE10A) inhibitor MR1916 reduced EtOH self-administration of high-drinking, post-dependent male rats, while the compound increased intake in females and in low-drinking males. MR1916 also increased the expression of medium spiny neurons (MSNs) markers Egr1, Fos, and FosB in the DLS, whose expression was modulated by sex and alcohol history. Finally, MR1916 elicited direct vs. indirect MSN markers differently in ethanol-naïve vs. post-dependent rats.
The topic of this research is timely as the phosphodiesterases (PDE) family of enzymes could represent a novel therapeutic target for alcohol use disorders. The findings are novel and interesting, the experiments rigorously performed and analyzed, and the manuscript is clearly written.
I only have a few minor comments that could improve the manuscript, which are listed down below.
- It would be helpful to the readers to better clarify the length of exposure to the ethanol vapors, and the timeline of the various experiments relative to the end of the vapor exposure.
- Please clarify the amount of RNA that was reverse transcribed.
- If the data shown in Figure 1 were first analyzed on both sexes together (with Sex as a factor), this should be reported (e.g. was an interaction Dose*Sex found?). However, we would understand if an interaction was not found, especially due to the high/low drinking levels differences.
- Do the authors have a possible explanation of the finding that Drd1 was increased in CIE rats and further increased by the inhibitor, in the DLS?
- Some additional discussion on the implications of the sex-related actions of the inhibitor would be helpful.
Reviewer 2 Report
Comments and Suggestions for Authors
The manuscript aims to determine if inhibiting phosphodiesterase 10A in striatal MSN impacts alcohol self-administration in rats. Biochemical measurements included gene expression changes in the dopamine and opioid pathways, as well as immediate-early genes in the striatum. This is an interesting approach towards reducing alcohol administration with the potential for translatability. The manuscript was thoughtfully written and does provide an interesting story. There biochemical findings in terms of sex-differences are the most interesting component of this manuscript. There are a few concerns with the analysis, which could be revisited to strengthen the paper.
Major
· The authors did justify why 0.6g/kg was chosen as a split between high and low drinking, and although this is interesting to examine, this left a small samples size per group. This is a bit of a concern for the ANOVA-based analysis approach, especially with regard to behavior. In Figure 1A, the low drinking males at .2umol/kg, this effect could very well be because of the potentially high intake outlier. The graphics do not convincingly show inverted-U effects in these rats or in the females as stated in the manuscript. It appears perhaps more like there is no effect in the lower drinking and females, but the data is just a bit noisy instead. If appropriate, it could be more interesting to show a more correlation-based approach between ethanol intake and the biochemistry rather than splitting groups.
· The discussion of the sex differences is very interesting. It would bolster the paper to expand this more to clinical implications, given the new and increasing interests in individual differences and sex differences in treatment approaches.
Minor
· It took me a while to understand the overall study design and experimental flow. It could be helpful to provide a graphical timeline for readers to review.
· It would be interesting to include information on if the inhibitor is normalizing drinking-related biochemical behavior.
· The vertical and horizontal lines on the graphs can be hard to differentiate with bars that are small.
Reviewer 3 Report
Comments and Suggestions for Authors
The present work resulted in a very interesting discovery regarding the role of PDE10A inhibitors in influencing postdependent alcohol consumption. The manuscript is well written. The results are clear. Bibliographic references are adequate. The figure should be presented in a larger format as some data becomes difficult to visualize.
Round 2
Reviewer 2 Report
Comments and Suggestions for Authors
The revised manuscript is very much improved with some new and interesting findings. The new version is almost ready for acceptance with just one concern. There authors mention inverted U-shape functions several times, but the corresponding figures do not appear to be an inverted U-shape function. Please clarify this.
Author Response
Thank you for raising this point to allow us to revise and clarify. The manuscript has been revised to change every mention of an “inverted U-shape function” to a “quadratic Dose contrast,” reflecting that the statistical model identified a significant quadratic relation of Dose to ethanol self-administration or preference (Lines 254, 255-257, 268, 272, 273, 529-531, 538-539, 573, 644-645). The revised manuscript further clarifies that this significant quadratic Dose contrast sometimes reflected that the low 0.05 µmol/kg dose increased intake or preference for the relevant subgroup as compared to vehicle (Lines 253-254, 275, 530, 645), while in other cases no significant pairwise differences vs. vehicle were observed (Lines 258-260).